


**Nine years of warming and nitrogen addition in the Tibetan grassland**
**promoted loss of soil organic carbon but did not alter the bulk change**
**of chemical structure**
Huimin Sun[1, 2], Michael W.I. Schmidt[2], Jintao Li[1], Jinquan Li[1], Xiang Liu[3], Nicholas
O.E. Ofiti[2], Shurong Zhou[4], Ming Nie[1*]
1. Ministry of Education Key Laboratory for Biodiversity Science and Ecological Engineering,
National Observations and Research Station for Wetland Ecosystems of the Yangtze Estuary,
Institute of Biodiversity Science and Institute of Eco-Chongming, School of Life Sciences, Fudan
University, Shanghai, 200433, China.
2. Department of Geography, University of Zurich, Zurich, Switzerland.
3. State Key Laboratory of Grassland Agro-Ecosystem, Institute of Innovation Ecology, Lanzhou
University, 222 Tianshui South Road, Lanzhou 730000, China.
4. Key Laboratory of Genetics and Germplasm Innovation of Tropical Special Forest Trees and
Ornamental Plants, Ministry of Education, College of Forestry, Hainan University, Haikou 570228,
China.
*Correspondence to*: Ming Nie (mnie@fudan.edu.cn)
**Abstract.** Understanding the changes in soil organic carbon (SOC) storage and
chemical stabilization dynamics is important for accurately predicting ecosystem C
sequestration and/or potential C loss, but the relevant information, especially for the
intervention of environmental controls on grassland soil is limited in Tibetan plateau
regions. Here we used a 9-year two-way factorial experiment involving warming with
open top chambers (+1.80 °C in the daytime and +0.77 °C in the nighttime at the soil





surface) and multilevel nitrogen (N) enrichment treatments (0, 5, 10, and 15 g m$^{-2}$ year$^{-1}$) in the Tibetan plateau to investigate the changes in SOC pool size and chemical
structure. 9-year warming treatment significantly decreased SOC stock in the Tibetan
grassland. We observed decreasing SOC concentrations which may be related to
changes in the C degrading enzymes. Surprisingly, the SOC molecular structure
remained unchanged in all N enrichment and warmed plots, suggesting that both
treatments had affected all forms of SOC, from simple and complex polymeric in a
similar way. Our results suggest that long-term warming stimulates soil C loss but no
preference in SOC loss with different chemical structure.
**Keywords:** global warming, nitrogen deposition, SOC, molecular structure, C
stabilization



## 1. Introduction

Soil organic matter is the largest organic carbon reservoir of near-surface terrestrial ecosystem (Dlamini et al. 2016). Even subtle acceleration in SOC decomposition will result in large $CO_2$ emissions (Davidson and Janssens 2006). So, knowledge of the factors affecting SOC storage and decomposition is essential for understanding the dynamically changing global C cycle. The influence of global warming on decomposition of soil carbon has been well documented (Poeplau et al. 2017, Guan et al. 2018, Ding et al. 2019b), but there remains considerable uncertainty in the potential response of soil C dynamics to the rapid global increase in reactive nitrogen (N, coming largely from agricultural fertilizers and fossil fuel combustion) as well as the combined effects with warming (Liang and Balser 2012, Devaraju et al. 2015, Li et al. 2017). The knowledge gap demonstrated a need to focus research on biological and physicochemical controls of SOC stabilization and destabilization processes as a basis for understanding causal relationships and key processes that determine pool sizes and turnover rates of functional SOC pools (von Lützow and Kögel-Knabner 2009).

Soil warming experiments in the field have shown that warming generates a considerable short-term soil C loss (Lu et al. 2013, Romero-Olivares et al. 2017). This loss declines over time (e.g. > 2 years) (Romero-Olivares et al. 2017), although there is evidence that it can continue for longer (e.g. > 20 years) (Melillo et al. 2017). Also, indirect effects of warming on nutrient cycling (Pendall et al. 2004) or plant inputs (Bradford et al. 2016) may have cascading effects on SOC quality and quantity (Lu et al. 2013) and consequently on microbial decomposition of SOC, including recent plant-



derived material (Hicks Pries et al. 2017) or older SOC (Vaughn and Torn 2019).
Because ecosystems in alpine meadow are normally N limited (Hobbie et al. 2002),
increased N released from decomposing SOC could stimulate plant productivity,
thereby increasing ecosystem C storage (Moscatelli et al. 2008). However, field
evidence suggests that soil microbial activity and biomass may also be N limited in
some C-rich ecosystems (Mack et al. 2004, Rinnan et al. 2007). Therefore, increased N
released from decomposition of SOC could further fuel microbial activity and decrease
soil C storage. Besides, according to the priming effect hypothesis, the increase in N
availability and labile C substrates promotes microbial C utilization, thereby increasing
the degradation of less decomposable SOC and leading to a negative effect on soil C
accumulation over the long term (Riggs and Hobbie 2016). However, it has been proven
difficult to quantify bulk SOC stocks changes and organic matter composition directly
(Sistla et al. 2013, Van Gestel et al. 2018). As alternatives, molecular-level techniques
can detect how temperature affects plant and soil organic matter, microbial growth and
their community composition under climate warming (Feng et al. 2008, Xue et al. 2016,
Pold et al. 2017).
Since the molecular structure of organic material has long been thought to determine
long-term decomposition rates in soil humic substances, solid-state CPMAS $^{13}$C NMR
spectroscopy has been successfully applied in studies on changes of SOC chemical
structure during organic matter decomposition without any physical or chemical
destruction (Schmidt et al. 2011). However, because of the large number of variables
affecting a spectrum, it is extremely difficult to obtain a complete and fine molecular



structure from a single spectrum without additional knowledge obtained by other
spectroscopic techniques (Ferrari et al. 2011). So, we employed another complementary
molecular-level analysis called diffuse reflectance infrared Fourier transform (DRIFT)
spectroscopy, which is a useful method for the characterization of organic matter (Olk
et al. 2000) and humic substances (Mao et al. 2008, Francioso et al. 2009), to explore
potential shifts in SOC composition in response to warming and N enrichment. The
structure of SOC could be very complex but by combining both techniques (DRIFT and
solid-state $^{13}$C NMR) complementary information could be obtained on aromatic and
aliphatic components (Ferrari et al. 2011).
Despite the importance of the response of SOC stocks to warming and N enrichment
in the intact ecosystem, this has not been assessed empirically in alpine meadows. This
knowledge gap is significant because the Tibetan Plateau stores a large C pool, with
36.6 Pg C stored in the top 3 m of the soil, accounting for 23.5% of China's total organic
soil-stored C and 2.5% of the global pool of soil C, which is of great importance in
regulating future global climate change and C emission (Genxu et al. 2002, Ding et al.
2019a). At the same time, the Tibetan Plateau has experienced climate warming at a
rate that is two times faster than that in other regions worldwide and is predicted to lead
to great soil C losses via microbial respiration in the future (Biskaborn et al. 2019). In
addition, during recent decades, the Tibetan Plateau has been subject to high levels of
N enrichment driven by agricultural activities (up to ~8.0 g m$^{-2}$ y$^{-1}$) (Gao et al. 2007,
Bo et al. 2012, Zhang and Fu 2020) and atmospheric N deposition (1 g N m$^{-2}$ y$^{-1}$) (Lü
and Tian 2007, Yu et al. 2019) with an annual rate of increase in deposition (0.053 g N



m$^{-2}$ y$^{-1}$) (Liu et al. 2013, Wang et al. 2019b), and this kind of enrichment has been shown
to induce soil C loss and affect SOC stabilization in this typical N-constrained
ecosystem (Xiao et al. 2021).
Since temperature is one of the main drivers of the vegetation growth and
decomposition of organic matter, on-going climate change may alter biophysical
processes with consequences for ecosystem functioning, especially in highly sensitive
cold regions such as the alpine meadow on the Tibetan plateau (Piao et al. 2006, Yang
et al. 2008). However, how and to what extent physical stabilization of SOC shifts may
occur, and consequently SOC storage and C-climate feedback would respond to
warming and N enrichment in an alpine meadow ecosystem, remains largely unknown.
Here, we used soils from a 9-year experiment with a two-way factorial design involving
soil warming (daytime: 1.80°C; nighttime: 0.77°C) and control plots and N enrichments
(0, 5, 10, and 15 g m$^{-2}$ y$^{-1}$; marked as N0, N1, N2, and N3, respectively) (Liu et al. 2016)
on the Tibetan Plateau to examine the changes in the stock and molecular structure of
SOC.




## 2. Materials and methods

### 2.1 Site description

Plot sampling was conducted in a grassland ecosystem located on the eastern edge

of the Tibetan Plateau, Maqu County, Gansu Province, China (101°53′ E, 35°58′ N,

3500 m above sea level, Figure 1 (NOAA 2015)), in August 2019. Alpine meadow is

the main vegetation type in this area. The area of alpine meadow accounts for more

than 44% of the area of alpine grasslands, and its SOC storage accounts for 56% of the

SOC storage of alpine grasslands on the whole TP (Yang et al. 2008). The soil in the

alpine meadow is classified as Mat-Cryic Cambisol (Hou et al. 2019). This region has

a typical plateau continental climate. The mean annual precipitation is 620 mm, and

most falls in the growing season (summer). The mean annual temperature is 1.2°C, with

the lowest monthly mean temperature occurring in January (−10.7°C) and the highest

monthly mean temperature occurring in July (11.7°C). During the past several decades,

the mean annual temperatures in the region have risen at a rate of 0.58°C per decade

(Liu et al. 2016). The plant community is dominated by perennial herbaceous species

of Poaceae, Ranunculaceae, and Asteraceae.

### 2.2 Experimental design and soil sampling

A field-based warming experiment was established in June 2011 with a split-plot

block design, in which both temperature (open-top chamber, +1.80°C in the daytime

and +0.77°C in the nighttime at the soil surface) and nitrogen (0, 5, 10, and 15 g m$^{-2}$ y$^{-1}$,

corresponding to N0, N1, N2, and N3, respectively) were manipulated, with six

replicates per treatment (Liu et al., 2016). The 48 plots with roughly the same species



diversity and community structure were 5 × 5 m and were separated by 1 m from
adjacent edges. Additional details can be found in our previous studies (Sun et al., 2023).
Surface layer (0-10 cm) soils were collected from these 48 plots using a 4-cm-diameter
auger in August 2019. Then, the fresh soil samples were transported to the laboratory
on ice.
**2.3 Soil analysis**
Soil microbial biomass carbon (MBC) was measured according to the chloroform
fumigation extraction method using a TOC analyzer (Multi N/C 3100, Analytik Jena
GmbH, Germany) (Vance et al. 1987). The soil pH was determined in a 1:5 soil: water
suspension with a pH meter (PHS-3D, Rex, Shanghai, China). For soil organic carbon
(SOC) analysis, air-dried soil was ground and HCl-fumigated (Komada et al. 2008),
and then the SOC concentration was determined with an elemental analyzer
(FlashSmart, Thermo Fisher Scientific, USA). The SOC stocks (0-10 cm) are calculated
by multiplying the SOC concentration by the bulk density (Walter et al. 2016). At each
site, all plants in three plots (50 × 50 cm) were harvested and dried to determine the
aboveground biomass.
We measured the activity of four extracellular enzymes in the soil at an in situ pH
(Nie et al. 2013). The absorbance of the C degradation enzymes β-D-cellubiosidase
(CB), α-glucosidase (AG), β-glucosidase (BG) and β-xylosidase (XYL) were measured
using a Tecan infinite M200 microplate fluorometer (Grodig, Austria) with 365 nm
excitation and 460 nm emission filters. The activities were expressed in units of nmol





h$^{-1}$ g$^{-1}$ dry soil. We combined CB, AG, BG and XYL into a C-degrading enzyme variable
(EnC).
**2.4 Bulk soil organic matter composition using DRIFT spectroscopy**

To characterize warming-induced changes in SOC composition, 6 mg of ground soil

sample was examined by diffuse reflectance infrared Fourier transform spectroscopy
(DRIFT). Mid-infrared spectra were recorded using a Bruker TENSOR 27 spectrometer
(Billerica, Massachusetts, USA) from 4000 to 400 cm$^{-1}$ (average of 16 scans per sample
at 4 cm$^{-1}$ resolution). Infrared absorption bands were represented by functional groups.
Additional details can be found in our previous studies (Ofiti et al. 2021).
**2.5 SOC molecular structure examination using NMR spectroscopy**

The soil samples used for NMR spectroscopy analysis were pretreated using HF (2%)

to eliminate paramagnetic materials, e.g. ferric ion and manganese ion, that may affect
the NMR signals (Skjemstad et al. 1994, Schmidt et al. 1997, Mathers et al. 2002). The
solid-state NMR spectra ($^{13}$C-CP-MAS) were recorded on a Bruker AVANCE III
600 MHz instrument (Bruker Instrument Inc., Billerica, MA, USA). The acquisition
conditions were set at frequency of 75.5 MHz, with 20 kHz spectra width, 5 kHz
spinning speed, 2 ms contact time, and 2.5 s recycle time. The regions of 0–210 ppm
spectra were plotted.

We examined seven chemical shift regions to represent the main C functional groups

(Golchin et al. 1997, Sun et al. 2019). We report proportions of each chemical shift area
and calculated 4 ratios indicative for the characteristics of soil organic matter. The alkyl
C, the most persistent fraction of SOC, comes from original plant biopolymers (such as





cutin, suberin and waxes) or from metabolic products of soil microorganisms (Ussiri
and Johnson 2003). As these materials decompose, the relative abundance of O-alkyl C
in the litter materials decreases, and there is a progressive increase in alkyl C
(Bonanomi et al. 2013). Therefore, the ratio of alkyl C to O-alkyl C (A/O-A = $C_{0-45}/C_{60-}$
$_{90}$) is an index represents the extent of SOC decomposition, the higher this ratio, the
higher the decomposition degree of SOC (Wang et al. 2015). Aromaticity ($C_{110-165}/C_{0-}$
$_{165}$), was used to indicate the complexity of molecular structure (Dai et al. 2001). The
ratio of aliphatic C/aromatic C (Alip/Arom), $C_{0-110}/C_{110-165}$, also indicates the molecular
structure of soil C, with higher Alip/Arom means less aromatic nuclear structure in
humus. The hydrophobic C/hydrophilic C (HB/HI) ratio, $(C_{0-45} + C_{110-165})/(C_{45-}$
$_{110} + C_{165-210})$, was used to reflect the stability of soil aggregation (Spaccini et al. 2006,
Wang et al. 2010).
**2.6 Regulating factors of SOC indicated by structural equation model**
To access the direct and indirect effects of external factors on SOC stock, structural
equation modeling (SEM) was performed using the R package 'plspm' and
'piecewiseSEM' (Li et al. 2020). For this purpose, firstly, all data were tested for
normality using the Kolmogorov–Smirnov test, and the non-normal variables were log-
transformed. Secondly, we established a prior model based on prior knowledge of
effects and relationships among the driving factors. Finally, we selected the best model
based on overall goodness of fits, including the chi-square ($\chi^2$) statistic, degrees of
freedom (df), whole-model $P$ value, goodness of fit index, and the root-mean-square
error of approximation (Schermelleh-Engel et al. 2003).



**2.7 Statistical analysis**

All data are presented as the mean values of six field replicates. Any significant differences in soil physicochemical properties among the different N enrichment levels and warming treatments were identified by using two-way ANOVA followed by Tukey's HSD post hoc test, with differences considered to be statistically significant at $P < 0.05$. The statistical analysis was conducted using SPSS 13.0 and R version 3.5.1 (R Foundation for Statistical Computing, Vienna, Austria, 2013).





## 3. Results

### 3.1 Bulk soil properties

Effects of N enrichment and warming treatment on soil properties were shown in the Figure 2 and Table S1. Warming aggravated N-induced soil acidification and microbial biomass C loss ($P < 0.05$, Figure 2a, 2e, 2f). Soil bulk density, SOC concentration, SOC stock, AGB and EnC increased significantly under N enrichment but decreased with N enrichment level as well as warming treatment ($P < 0.05$, Figure 2, Table S1, Figure S1). Both N enrichment and warming significantly decreased C/N ratio ($P < 0.05$, Figure 2d).

### 3.2 SOC speciation as seen by DRIFT and NMR spectroscopy

Changes in SOC molecular composition became apparent in diffuse reflectance infrared Fourier transform (DRIFT) and nuclear magnetic resonance (NMR) spectra (Figure 3, 4 and Figure S2, S3). In all N enrichment and warming treatments, there was a statistically non-significant change in the SOC composition and molecular structure observed by both DRIFT and NMR spectra. The relative abundance of carbonyl/carboxyl C=O, and C=C aromatics compounds as well as lignin-like residues decreased (non-significant) after N enrichment, however, kept steady in warming plots (Figure 3 and S2).

The results of $^{13}$C NMR spectroscopy indicated the relative abundance of different C components (Table 1, Figure 4 and S3), showing that the proportion of the seven C functional groups did not change in soils under N enrichment and warming treatments. The relative proportions of the seven C functional groups were similar in the following





abundance order: O-alkyl C (mean 33%), followed by alkyl C (mean 22%), aromatic C
(mean 12%), N-alkyl C (11%), carbonyl C and di-O-alkyl C (mean 8%), and finally
phenolic C (mean 3%). The four indexes which can represent the extent of SOC
decomposition observed by NMR spectra also showed no significant difference under
all the treatments (Figure 4), suggesting that SOC showed a similar degradation state at
all N level enrichments and warming treatments.
**3.3 Factors driving the SOC stocks**
We then developed a structural equation model (SEM) to assess the direct and
indirect effects of soil variables on the SOC stocks (Figure 5). The SEM results revealed
strong connections among global change, biotic, and edaphic factors (Figure 5),
demonstrating a need to consider their interactions when predicting SOC stock and its
response to N enrichment and warming. Overall, the SEM explained 44% and 55% of
the variance in SOC stock driven by N enrichment and warming treatment, respectively.
In both patterns, C-degrading enzymes showed an important indirect factor in
regulating SOC stock. N enrichment had a positive effect on SOC stock by enhancing
enzyme activities. In contrast, warming had a negative effect on SOC stock by
inhibiting microbial enzymes. Besides, warming had a strong negative direct effect on
SOC stock (Figure 5b).



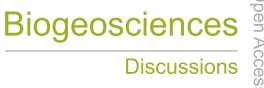

## 4. Discussion

### 4.1. Effects of warming and N enrichment on soil C pool size

It is suggested that small N inputs can decrease $CO_2$ emissions by changing the interaction between plants and soil microbes in N-limited ecosystems, for example, by increasing plant productivity and root biomass and then organic C inputs to the soil by promoting N availability and thus retard litter and SOC decomposition (Franklin et al. 2003, Mo et al. 2008, Zhou et al. 2014). However, in an alpine grassland, Jiang et al. found that both plant growth and microbial activity were generally N-limited, but the ability of plants to capture soil inorganic N was much stronger than that of soil microorganisms. When N was added, increased N availability resulted in increased plant growth, microbial activity and plant biomass (Micks et al. 2004). Therefore, the decomposition of litter and SOM is enhanced by increasing the quantity of litter input or by elevating microbial activity, and consequently, soil functions would shift from C sequestration to C loss. The increased N has consequently reduced the soil pH by 0.26 globally in only one decade, which may significantly influence the microbial community composition and activity and then SOC sequestration capacity (Geisseler and Scow 2014, Tian and Niu 2015, Raza et al. 2021). This speculation is consistent with our results that N input below the critical level may be beneficial for C sequestration in alpine meadows of the TP and can partly explain the patterns of SOC pool size under various N enrichment levels in this study.

In our study presented here, the 9 years of warming resulted in a very significant SOC loss of 14 to – 28 % (Figure 2 and S1). The Tibetan plateau stored large amounts





of SOC because of the permafrost soil, where limited C decomposition has led to the
accumulation of large SOC stocks (Hengl et al. 2014, Schuur et al. 2015). Previous
studies showed that the vulnerability of soils with large C stocks derives from the high
temperature sensitivity of C decomposition and biogeochemical restrictions on the
processes driving soil C inputs. Contrast with that, in soils with low initial C stocks,
small losses coming from accelerated decomposition induced by rising temperature
may be offset by concurrent increases in plant growth and soil C stabilization (Day et
al. 2008, Macias-Fauria et al. 2012, Crowther et al. 2015). However, in areas with larger
SOC stocks, accelerated decomposition exceeds the potential C accumulation of plant
growth, contributing to a significant C loss to the atmosphere.
**4.2. Effects of N enrichment and warming on SOC chemical compositions**

SOC chemical composition not only is controlled by the chemistry of the plant

materials input to the soil, but also by the microbial processing and degradation of SOC
(Baldock et al. 1992). Although N addition can stimulate plant growth and increase
litter fall, it can also accelerate or slow down microbial processing of plant residues,
thus altering the chemical composition of SOC (Wang et al. 2019a). Surprisingly we
observed that the SOC molecular structure remained unchanged in all N enrichment
and warmed plots (Figure 3, 4).

As the predominant chemical component of SOC across all treatments in our study,

O-alkyl C is mainly composed of carbohydrates, peptides and other labile organic
components derived from the fresh material, which could be preferentially degraded
compared with more resistant components such as alkyl C (Simpson and Simpson, 2012,





He et al. 2018). So, we hypothesized that this result could indicate that N and warming
may have the same impact on the input of fresh plants on the Tibetan plateau. Unlike
O-alkyl C, lipids represent the main source of alkyl C (aliphatic chains), which is
derived from original plant biopolymers. Lignin and tannin represent the main source
of aromatic C, together with phenolic C, mainly originating from lignin and amino acids
of peptides (Baldock et al 1992). All these components are more resistant to microbial
decomposition than labile O-alkyl C (Simpson and Simpson, 2012). Our results
suggested that the proportions of the stable SOC chemical structures remained the same
between the treatments, indicating the synchronous degradation of SOC. The alkyl/O-
alkyl ratio and aromaticity, normally regarded as the indicator of the relative stage of
SOC degradation and has been widely used as an indicator to reflect the complexity of
SOC chemical structure (Baldock et al. 1992), exhibited no change after 9-year N
enrichment and warming, suggesting that all N levels and warming treatments exerted
similar effects on the degradation of SOC and aromatic and complex molecular
structure (Zhang et al. 2013).
Infrared spectroscopy of SOC showed a statistically non-significant change in the
treated plots and the control plots, which were consistent with a previous study that
showed the 4.5 years of +4 °C whole-soil warming did not change the relative
abundance of carbonyl/carboxyl C=O, and C=C aromatics compounds in the surface
soils (above 20 cm) from a forest (Ofiti et al. 2021). Collectively, the above results
suggested that molecular structure of surface SOC may not be as sensitive to long-term
warming as we thought before (Atanassova and Doerr 2011, Chen et al. 2018a). Surface



SOC is dominated by recent (less transformed) plant-litter inputs, which is less
degraded and transformed than subsoil SOC (Ofiti et al. 2021). The lack of change in
plant- and microorganism-derived organic matter in the surface soil may be due to slight
drying and warming near the surface (Soong et al. 2021) which could have inhibited or
resulted in relatively less surface inputs. We noticed that warming significantly reduced
aboveground biomass in this study (Figure 2). Compared with labile SOC, stable SOC
can be more vulnerable to priming once microbes are provided with exogenous C
substrates. This high vulnerability of stable SOC to priming warrants more attention in
future studies on SOC cycling and global change (Zhang et al. 2022). Overall, stable
functional SOC molecular structure indicated that soil warming and N enrichment had
similarly affected easily decomposable and stabilized SOC of this C-rich grassland soil
despite the C loss.
**4.3 Regulating factors of SOC stock**
Our interpretation that prolonged warming could reduce SOC storage is further
supported by the simultaneous reduction of different C pool sizes characterized with
various chemical structural complexity with long-term warming. Many previous studies
have shown that microorganisms preferentially use the labile C pool for community
utilization and turnover after short-term warming (Melillo et al. 2002, Kirschbaum
2004). However, after the initial microbial assimilation of readily accessible SOC with
warming, soil microorganisms can acclimate to C starvation through utilization of
chemical less available C with continued warming (Chen et al. 2020). This
transformation in microbial preference of C substrates can be facilitated by changes in



C-degrading enzyme activities (Crowther and Bradford 2013).
Our results indicated that C-degrading enzymes could play a key role in regulating
soil C storage, which is in line with previous explanations for continued soil C loss with
long-term warming, such as shifts in microbial community and physiology (Melillo et
al. 2017, Metcalfe 2017), changes in microbial carbon use efficiency (Tucker et al.
2013), and increased microbial accessibility to litter and SOC (Doetterl et al. 2015,
Bailey et al. 2019), which are all closely related to changes in microbial C-degrading
enzyme activities. For example, warming decreased the abundance of lignin-derived
compounds but increased ligninase activity in a mixed temperate forest (Feng et al.
2008). Although only cellulase activity was measured in our study, a previous meta-
analysis study has shown significantly increasing ligninase activity after warming,
enhancing the evidence of microbial response to recalcitrant C pools and the evidence
of simultaneous loss of different C fractions after long-term warming (Chen et al.
2018b). Microbial utilization of recalcitrant C pools could substantially accelerate
overall soil C loss because depolymerization of these recalcitrant macromolecules
increases microbial accessibility to litter and SOC previously protected by recalcitrant
C pools (Schmidt et al. 2011, Lehmann and Kleber 2015, Paustian et al. 2016).




**5. Conclusion**

Based on a 9-year warming (+1.80 °C in the daytime and +0.77 °C in the nighttime at the soil surface) and different level N enrichment experiment (0, 5, 10, and 15 g m$^{-2}$ year$^{-1}$), we examined the responses of SOC stocks and their molecular components in a Tibetan alpine meadow ecosystem. In summary, our results show little effects of soil warming and N enrichment on the chemical composition of bulk soil despite ongoing C loss in the warmed plots of the study site (Figure 2). The SOC molecular structure suggested that the easily decomposable and stabilized SOC are similarly affected after 9-year warming and N treatments despite the large changes in SOC stocks. Given the long residence time of some SOC (Schmidt et al., 2011), the similar loss of all measurable chemical forms of SOC under global change treatments could have important climate consequences. Permafrost soils contain half of global SOC stocks (Ding et al. 2016, Hugelius et al. 2020). While we found little effects of soil warming on SOC chemistry and molecular structure of bulk soil, consistent and long lasting changes could appear with prolonged soil warming and decreasing SOC stocks in the following years.



**Data availability**

The data that support the findings of this study and those not presented within the article and its Supplementary Information file are available from https://doi.org/ 10.5281/zenodo.8289311.

**Author contributions**

M.N. developed the original ideas presented in the manuscript; H.S. performed the overall analysis with the assistance from Jintao Li. and Jinquan Li, N. O.; X.L. and S.Z. organized the field experiment; H.S., M.S. and M.N. wrote the first draft, and all authors jointly revised the manuscript.

**Competing interests**

The contact author has declared that none of the authors has any competing interests.

**Acknowledgements**

This work was supported by the National Natural Science Foundation of China (91951112 and 32101377), the Program of Shanghai Academic/Technology Research Leader (21XD1420700), the'Shuguang Program' supported by Shanghai Education Development Foundation and Shanghai Municipal Education Commission (21SG02), the Shanghai Pilot Program for Basic Research—Fudan University 21TQ1400100 (21TQ004), the Science and Technology Department of Shanghai (21DZ1201902), the Shanghai Pujiang Program (2020PJD003), and the Swiss National Science Foundtion SNF-project 172744 (DEEP C) awarded to M.W.I.S. and N.O.E.O..



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

724



**Figure and table legends**

**Figure 1.** Location of the studied sites.

**Figure 2.** N and warming-induced changes in the soil properties (mean ± SE, n = 6). Control (white bar) and warmed plots (black bar) at four different levels of simulated N deposition. N0, N1, N2, and N3 indicate N-enrichments of 0, 5, 10, and 15 g N m$^{-2}$ year$^{-1}$, respectively. Parameters are: Soil pH (a); AGB, aboveground biomass (b); EnC, C-degrading enzymes (c); C/N, ratio of soil C concentration to N concentration (d); MBC, microbial biomass carbo (e); SOC, soil organic carbon stock (f).

**Figure 3.** N and warming-induced changes in the relative abundance of different functional groups identifiable by diffuse reflectance infrared Fourier transform (DRIFT) spectroscopy in warmed and control plots (mean ± SE, n = 6). The spectral regions were assigned to aromatic carbonyl/carboxyl C=O groups, aromatic C=C groups, lignin-like residues, and cellulose/phenolic. No significant differences were found.

**Figure 4.** our different SOC chemical structural complexity indexes (mean ± SE, n = 6) from solid-state $^{13}$C CPMASNMR spectra of soil samples from different treatments. A/O-A=Alkyl C/O-alkyl C; HB/HI = hydrophobic C/hydrophilic C; Alip/Arom = aliphatic C/aromatic C. No significant differences were found.

**Figure 5.** he factors regulating the SOC stock under (a) N enrichment and (b) warming treatment. In the structural equation model (SEM) analysis, black arrows represent significant positive pathways, gray arrows represent significant negative pathways, and gray dashed arrows indicate nonsignificant pathways. Values next to the arrows represent standardized effect sizes with statistical significance (*$P < 0.05$; **$P < 0.01$;





***$P < 0.001$). The thickness of the arrow represents the standardized effect sizes. C-
degrading enzyme indicate sum of β-D-cellubiosidase (CB), α-glucosidase (AG), β-
glucosidase (BG) and β-xylosidase (XYL). Goodness-of-fit statistics for the model are
shown as follows: (a), $\chi^2 = 4.53$, $P = 0.53$, GFI =0.99, RMSEA < 0.001; (b), $\chi^2 = 4.47$,
$P = 0.486$, GFI = 0.99, RMSEA < 0.001.
**Table 1.** Relative intensities (mean ± SE, n = 6) of different carbon chemical shifts from
solid-state $^{13}$C CPMAS NMR spectra of soil samples from N and warming treatments.
No significant differences were found.





**Figure 1**

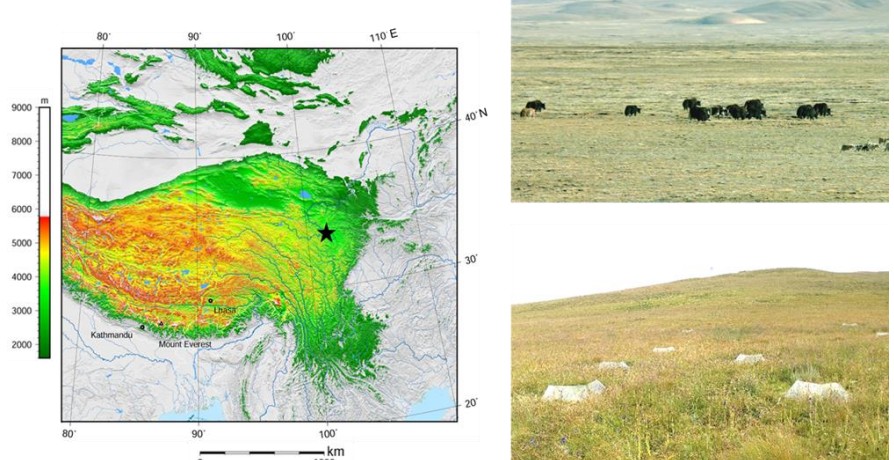


Note: the map was cited from the Wikimedia Commons website (Tibet and
surrounding areas above 1600m, created using the Generic Mapping Tools)



**Figure 2**

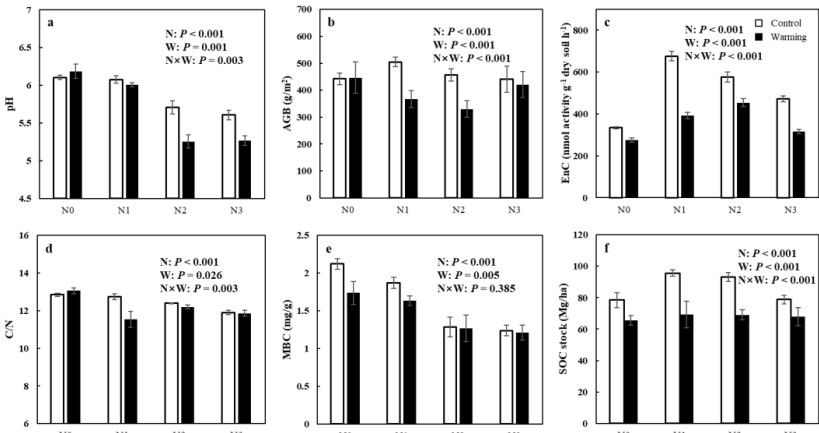








**Figure 3**

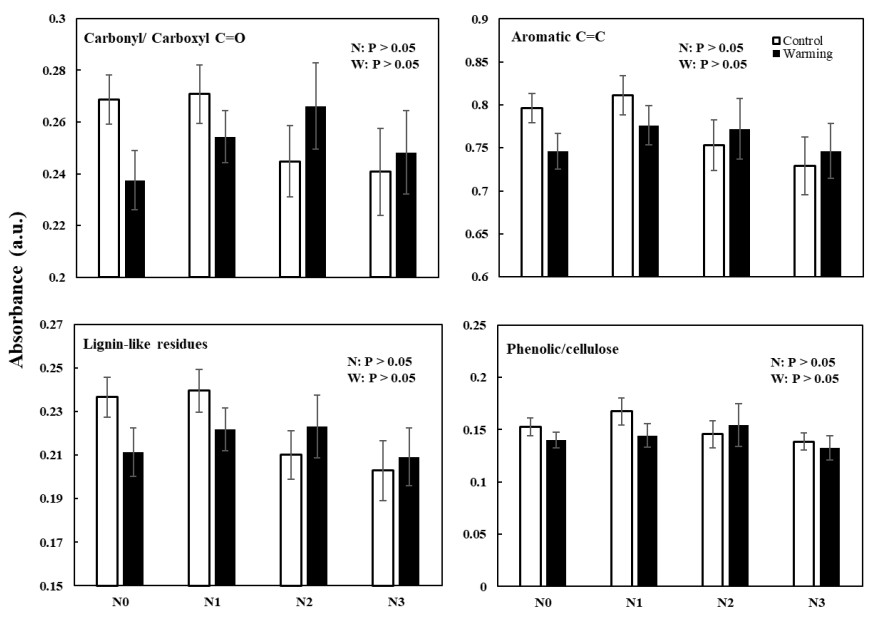




**Figure 4**

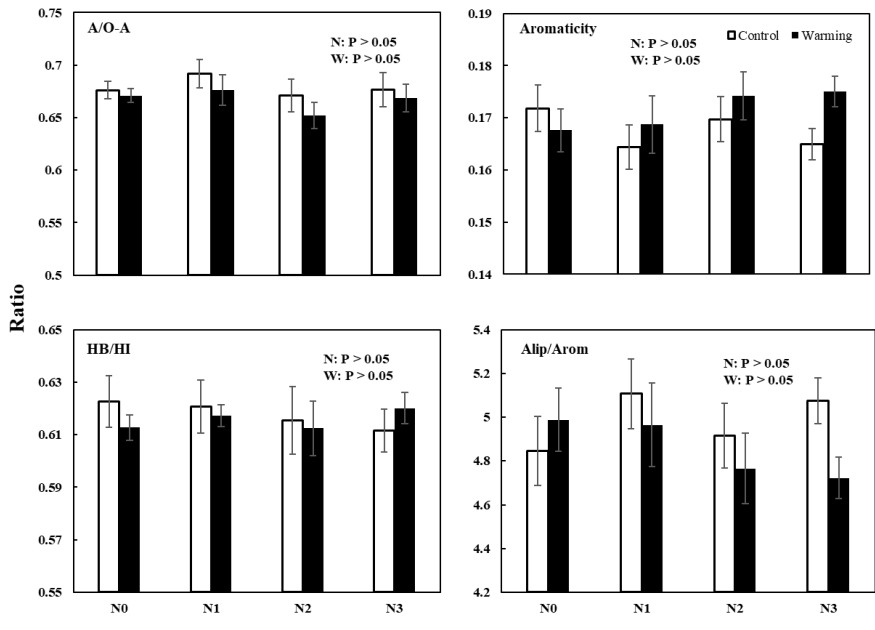




**Figure 5**

(a)

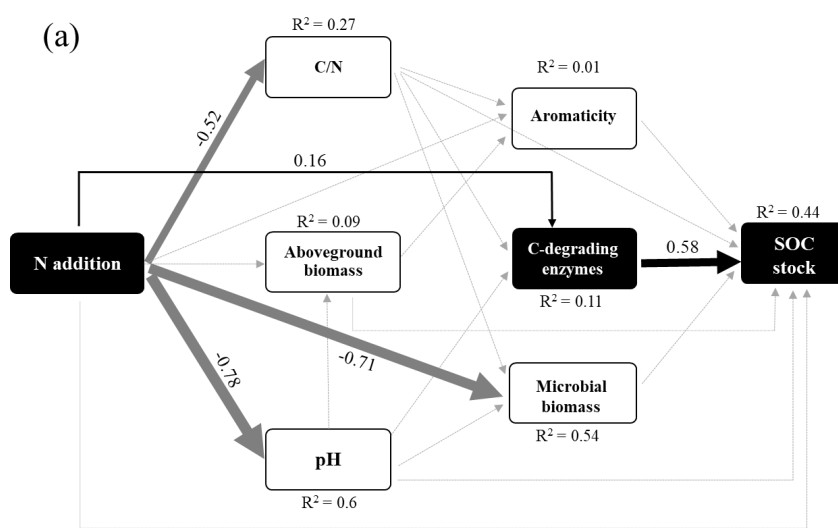


(b)

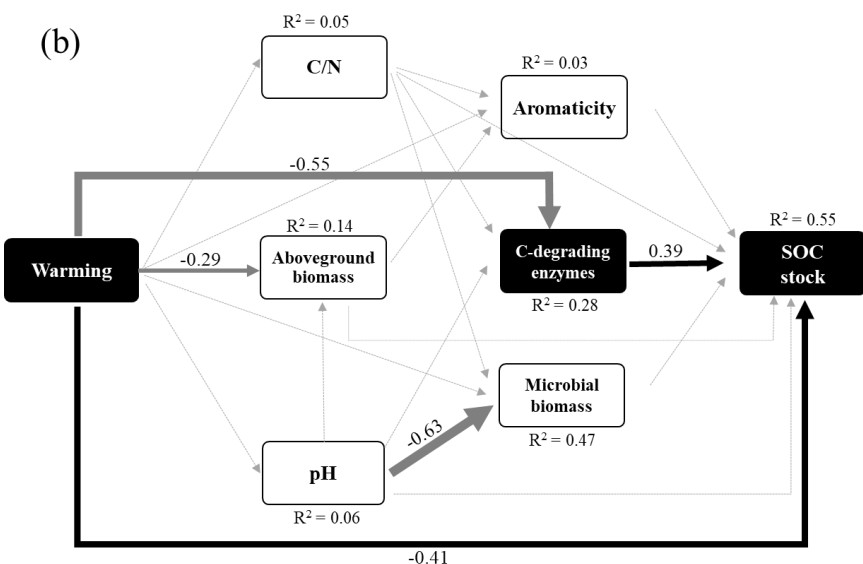






**Table 1**

| Chemical shifts | | Treatment | | | |
|---|---|---|---|---|---|
| | | N0 | N1 | N2 | N3 |
| **$^{13}$C NMR (%)** | Alkyl C (0-45 ppm) C | 22.64±0.15 | 23.22±0.17 | 22.55±0.2 | 22.85±0.32 |
| | W | 22.68±0.24 | 22.75±0.51 | 22.07±0.26 | 22.29±0.29 |
| | N-alkyl C (45-60 ppm) C | 10.91±0.09 | 11.08±0.11 | 10.82±0.08 | 10.92±0.1 |
| | W | 10.73±0.06 | 10.6±0.24 | 10.54±0.08 | 10.67±0.16 |
| | O-alkyl C (60-90 ppm) C | 33.5±0.35 | 33.62±0.53 | 33.67±0.54 | 33.83±0.47 |
| | W | 33.8±0.3 | 33.65±0.23 | 33.88±0.45 | 33.36±0.31 |
| | di-O-alkyl C (90-110 ppm) C | 8.74±0.09 | 8.65±0.09 | 8.9±0.13 | 8.82±0.12 |
| | W | 8.88±0.09 | 9.02±0.11 | 8.88±0.1 | 9.02±0.26 |
| | Aromatic C (110-145 ppm) C | 12.25±0.33 | 11.62±0.26 | 12.05±0.3 | 11.45±0.28 |
| | W | 11.86±0.36 | 12.1±0.45 | 12.34±0.38 | 12.44±0.21 |
| | Phenolic C (145-165 ppm) C | 3.47±0.1 | 3.44±0.11 | 3.47±0.08 | 3.64±0.05 |
| | W | 3.46±0.06 | 3.32±0.05 | 3.56±0.07 | 3.54±0.09 |
| | Carbonyl C (165-210 ppm) C | 8.49±0.1 | 8.37±0.17 | 8.54±0.18 | 8.49±0.3 |
| | W | 8.61±0.16 | 8.57±0.24 | 8.73±0.14 | 8.68±0.19 |
