# Peer review of "Nine years of warming and nitrogen addition in the Tibetan grassland"

_Biogeosciences, 2023_

## Author Comment (AC2)

Dear Editor and Reviewers,
Thank you for dedicating your time to review our manuscript and providing us with valuable feedback. We are grateful for the positive comments highlighting the potential significance and interest of our study. We highly value all the critical comments, as they have greatly contributed to the improvement of our work.
In response to the reviewer' comments, we have thoroughly revised the manuscript. We have carefully considered each comment and incorporated the necessary changes and refinements throughout the revised manuscript. To facilitate your review, we have provided detailed responses to each comment in blue color below. Additionally, you can refer to the tracked changes in the manuscript for a comprehensive overview of the revisions made.
Once again, we sincerely appreciate your time and expertise in evaluating our work, and we hope that the revisions have strengthened the manuscript in terms of clarity, accuracy, and overall quality.

Best Regards,
Huimin Sun on behalf of all co-authors.

**Response to RC#2**

Warming and nitrogen deposition are two important topics in current research. Understanding the changes in soil organic carbon (SOC) storage and chemical stabilization dynamics is important for accurately predicting ecosystem C sequestration under climate change. This study showed that little effects of soil warming and N enrichment on the chemical composition of bulk soil despite ongoing C loss in the warmed plots. Although these findings have significant implications for improving the climate-C model, this article needs to be revised in multiple places.

1) The description of the result needs to be accurate. For example, there were many references to increase or decrease, and the object of comparison should be clearly defined. According to results of DRIFT and solid-state 13C NMR, warming did not have a significant effect on the soil organic matter composition. However, the authors reported that the easily decomposable and stabilized SOC are similarly affected after 9-year warming and N treatments. "Similarly" is ambiguous here and needs to be clear.

Thank you for your review. The Results and Discussion section has been completely corrected and rewritten after revising all comments from both reviewers. Unclear statements in the text have been rewritten. For example:
'The relative abundance of carbonyl/carboxyl C=O, C=C aromatics compounds as well as lignin-like residues decreased slightly after N enrichment, not significantly though. The relative abundance of the phenolic/cellulose remained stable in all treatments (Figure 3 and S2).'

2) The importance of C-degrading enzymes under warming in regulating SOC was emphasized, but it was understated in the introduction, results, and conclusion.

Done as suggested. The importance and potential mechanisms of C-degrading enzymes have been added into the manuscript. For example:
'Soil N availability would strongly affect microbial physiology and C-degrading enzymes (EnC), which can subsequently alter soil C feedbacks to warming (Mack et al. 2004; Contosta & Cooper 2015). EnC has been shown to play an important role in SOM nutrient cycling and catabolism (Chen et al. 2018a), and information on such activity can be used to investigate substrate nutrient demand and response to environmental changes (Allison et al. 2010; Wang et al. 2015).'

3) The alkyl/O-alkyl ratio and aromaticity, normally regarded as the indicator of the relative stage of SOC degradation and has been widely used as an indicator to reflect the complexity of SOC chemical structure. Why is it that only the aromaticity was considered in the SEM?
Because these two metrics are identical in indicative meaning and are strongly correlated. Aromaticity was chosen because it is more widely used and recognized.

4) Soil moisture change under warming is a key index that affects SOC dynamics. Does warming affect soil moisture change?
Yes, warming do affect soil moisture. But in this study, we focused more on the physical changes in the chemical structure and stabilization of the carbon pool. We were not able to take into account all the changes in soil properties. Thank you for your suggestion and we will consider soil moisture in our research in the future.

5) Nitrogen addition is also an important environmental factor, but the study focused more on warming.
Thanks for your suggestion. The manuscript has been completely rewritten after revising all comments from both reviewers. The content and importance of nitrogen addition was expanded and more fully described.

6) Treatment time seems to be a very important factor because it was used in the title, but the discussion did not involve time.
In fact, the discussion section mentions the issue of time several times, just not in terms of specific treatment duration as a direct description, but more in terms of comparing the potential difference between short-term warming and long-term warming. For example:
'Many previous studies have shown that microorganisms preferentially use the labile C pool for community utilization and turnover after short-term warming (Melillo et al. 2002, Kirschbaum 2004). However, after the initial microbial assimilation of readily accessible SOC with warming, soil microorganisms can acclimate to C starvation through utilization of chemical less available C with continued warming (Chen et al. 2020). This transformation in microbial preference of C substrates can be facilitated by changes in C-degrading enzyme activities (Crowther and Bradford 2013).'
What this study would like to highlight is that this was a long-term warming experiment of up to 9 years, and the chemical structure of the organic carbon was not significantly

altered compared to the short-term experiment. In the future we should focus on changes in soil carbon pool turnover under relatively long-term treatments, which may be more meaningful than results from short-term treatments.

7) Suggest adding hypotheses.
Done as suggested. The hypothesis has been added at the end of the Introduction section as follows:
'We hypothesized that 9-years N enrichment and warming would affect SOC stock and the chemical structure of the SOC. N enrichment below a certain threshold may favor C sequestration in the alpine grassland ecosystem but warming may result in the C loss. And added N would stimulate hydrolytic enzyme activity while warming would repress enzyme activity. Finally, we hypothesized that variation in enzyme response to N and temperature would emerge as an important explanation for variability in the effect of added N and warming on SOC stock.'

Specific comments:

In the abstract, the background of warming and nitrogen deposition should be added. Based on SEM, it is clear that nitrogen addition and warming affect SOC in inconsistent ways. In addition, at the end, the significance of the results of this study needs to be emphasized.
Done as suggested.  The beginning of the Abstract has been rewritten as follows:

'Nitrogen (N) and warming effects on ecosystem carbon (C) budgets and stabilization are critical to understand as C sequestration is considered as a mechanism to offset anthropogenic CO2 emissions, which is important for accurately predicting ecosystem C sequestration and/or potential C loss, remaining controversial though. But the relevant information, especially for the intervention of environmental controls on grassland soil is limited in Tibetan plateau (TP) regions.'

L88-89: Inaccurately, there have been many studies focusing on the effect of warming and nitrogen enrichment on SOC stocks in alpine meadows.
Done as suggested. The text has been rewritten as follows:
'Despite the importance of the response of SOC stocks to warming and N enrichment in the intact ecosystem, results about the chemical stabilization mechanisms (i.e. molecular structure of SOC) in alpine meadows remained controversial.'

L108: This study does not involve the physical stability of SOC.
Done as suggested. The text has been corrected to 'chemical stabilization'.

L125: Please specify the full name of TP.
Done as suggested. The abbreviations have been checked carefully.

L139: How the 48 plots are calculated, it is recommended to list all the treatments.

Done as suggested. The sentence has been rewritten: 'The 48 plots (8 treatments (N0, N1, N2, N3, WN0, WN1, WN2, WN3) with 6 replicates each treatment) with roughly the same species diversity and community structure were 5 × 5 m and were separated by 1 m from adjacent edges.'

L142-143: How much soil was taken from each plot and how to ensure the uniformity of sampling.

Five cutting rings (100cm3) of soil were taken from each sample plot and brought back to the laboratory. All samples were homogenized after removing stones and rhizomes before starting soil analysis.

L154-155: What does 'each site' refer to here?

Thanks for your suggestion. The text has been corrected to 'At these 48 sites'.

L164: Does it mean that only the samples treated with warming were measured?

Thanks for your suggestion. The text has been corrected to 'warming/nitrogen-induced changes in SOC composition'.

L169: DRIFT is a key technique in this study, and it is recommended to write in detail.

Done as suggested. The section has been added in detail: 'Infrared absorption bands were represented by functional groups as follows: aliphatic C–H (2900 cm$^{-1}$), aromatic esters, carbonyl/carboxyl C– –O (1735–1720 cm$^{-1}$), aromatic C– –C (1660–1600 cm$^{-1}$, 1430–1380 cm$^{-1}$), lignin-like residues (1515–1500 cm$^{-1}$), phenolic/cellulose (1260–1210 cm$^{-1}$), and aromatic C–H (880, 805, 745 cm$^{-1}$) carbon (Niemeyer et al. 1992; Leifeld, 2006; Chatterjee et al. 2012). A summary of the absorption bands associated with different compound classes can be found in Figure 3.'

L196-204: Suggest placing it in the following Statistical analysis.

Done as suggested. The section has been moved to the 'Statistical analysis'.

L215-216: Suggest deletion.

Done as suggested.

L216-217: N enrichment and warming have significant interaction on pH, and not all warming aggravated N-induced soil acidification.

Done as suggested. This section has been corrected and rewritten.

L217-219: Similarly, there were significant interactions on SOC concentration, SOC stock, AGB, EnC and C/N. Different treatments have to be analyzed separately, rather than looking directly at the main effect. Here is an increase or a decrease described. What are they referring to?

This section has been completely rewritten to reduce ambiguity:

'N enrichment and warming have significant interaction on pH, AGB, EnC, C/N and

SOC stock (P < 0.05, Figure 2). Soil bulk density, SOC concentration, SOC stock, AGB and EnC increased significantly under N enrichment but decreased with N enrichment level as well as warming treatment (P < 0.05, Figure 2, Table S1, Figure S1). Warming exacerbates soil acidification and decreases in C-degrading enzyme activities and SOC stocks caused by high-level N concentrations (Figure 2a, 2c, 2f). Both N enrichment and warming significantly decreased C/N ratio (P < 0.05, Figure 2d).'

L225: It is recommended to describe the results of the interaction first and then the results of the main effect. Similarly, what is the reference object that is increased or decreased?

Done as suggested. This section has been rewritten. The interaction effects were described at first and then the individual effects were followed.

'N enrichment and warming have significant interaction on pH, AGB, EnC, C/N and SOC stock (P < 0.05, Figure 2). Soil bulk density, SOC concentration, SOC stock, AGB and EnC increased significantly under N enrichment but decreased with N enrichment level as well as warming treatment (P < 0.05, Figure 2, Table S1, Figure S1). Warming exacerbates soil acidification and decreases in C-degrading enzyme activities and SOC stocks caused by high-level N concentrations (Figure 2a, 2c, 2f). Both N enrichment and warming significantly decreased C/N ratio (P < 0.05, Figure 2d).'

L234, L237-238: What treatments are similar here? Please supplement the relevant statistics, the current statistics could not see this result.

Done as suggested. The sentence has been rewritten as follows:

'The relative proportions of the seven C functional groups were similar in the 8 treatments in the following abundance order: O-alkyl C (mean 33%), followed by alkyl C (mean 22%), aromatic C (mean 12%), N-alkyl C (11%), carbonyl C and di-O-alkyl C (mean 8%), and finally phenolic C (mean 3%) (Table 1, Figure S3).'

L270-271: Suggest changing to a hypothesis and placing it in the Introduction. Furthermore, what does the critical level refer to here? Please specify the patterns of SOC pool size under various N enrichment levels.

Done as suggested. The hypothesis has been added at the end of the Introduction section as follows:

We hypothesized that 9-years N enrichment and warming would affect SOC stock and the chemical structure of the SOC. N enrichment below a certain threshold may favor C sequestration in the alpine grassland ecosystem but warming may result in the C loss. And added N would stimulate hydrolytic enzyme activity while warming would repress enzyme activity. Finally, we hypothesized that variation in enzyme response to N and temperature would emerge as an important explanation for variability in the effect of added N and warming on SOC stock.

And this sentence has been rewritten as follows:

'This speculation is consistent with our results that N input below the critical level (for example, 10 g m$^{-2}$ in this study) may be beneficial for C sequestration in alpine meadows of the TP and can partly explain the patterns of SOC pool size under various

N enrichment levels in this study. Specifically, the SOC stock increased following N enrichment, but as the N addition concentration increased, this growth progressively diminished, eventually even disappearing. Our results revealed that alpine grassland ecosystems on the TP may become a potential C source under future scenarios of increasing N enrichment.'

L277-278: Please provide references.
Done as suggested. The references have been added:
'Previous studies showed that the vulnerability of soils with large C stocks derives from the high temperature sensitivity of C decomposition and biogeochemical restrictions on the processes driving soil C inputs (Davidson and Janssens 2006; García-Palacios et al. 2021).'

L284-285: Please provide relevant data support.
Done as suggested. The references have been added:
However, in areas with larger SOC stocks, accelerated decomposition exceeds the potential C accumulation of plant growth, contributing to a significant C loss to the atmosphere (Luo et al. 2019).

L299: Obviously, this conjecture is not applicable in this study, as the interaction between warming and nitrogen addition has a significant effect on the aboveground biomass. Why compare the effects of nitrogen addition and warming, rather than comparing the differences in nitrogen addition or warming gradients?
We arrived at this speculation because no significant differences in the chemical structure of SOC were observed at all gradients of nitrogen addition as well as at the warming treatments. Although the interactive effects of nitrogen addition and warming significantly increased aboveground biomass, the accelerated decomposition may have counteracted the carbon sequestration of plant growth, which is not inconsistent with the results of carbon loss.

L306: What does the 'treatments' here refer to?
Sorry for the unclear description. The content has been changed to the following:
'the different N enrichment concentrations and warming treatments'

L309: This study is not a comparison of 9 years before and after N enrichment and warming, so the term "after" is not appropriate, and it is suggested to change to "exhibited no significant difference among N enrichment or warming treatments".
Done as suggested.

L310-311: Again, what is the purpose of describing that all N levels and warming treatments show similar effects?
We would like to convey the conclusion of this study. To be specific, stable functional SOC molecular structure indicated that soil warming and N enrichment had similarly affected easily decomposable and stabilized SOC of this C-rich grassland soil despite

the C loss.

L324-325: The description needs to be accurate. Not all treatments show a significant decrease in aboveground biomass with warming.
This sentence has been corrected: 'We noticed that warming significantly reduced AGB under N1 and N2 enrichments in this study (Figure 2).'

L330: It's very confusing, how did you get this result?
This sentence has been rewritten to avoid excessive speculation on the conclusion:
'Overall, the stabilized functional SOC molecular structure suggests that soil warming and N enrichment had similarly affected the labile and stabilized SOC of this C-rich grassland soil at the level of chemical stability of organic C molecules, along with the C loss.'

L332: Part 4.3 only considers the regulation of SOC by extracellular enzymes under warming. What is the mechanism of regulating SOC changes under nitrogen addition?
The 4.3 section has been completely rewritten to add potential mechanisms for nitrogen addition on enzyme activity as well as organic carbon availability. The following content has been added:
'While N fertilization exerts both direct and indirect impacts on SOC, its influence on carbonates is direct, leading to continuous losses. This not only serves as a source of atmospheric $CO_2$ (Kim et al., 2020; Raza et al., 2020; Zamanian et al., 2018) but also degrades soil structure and affects physical, chemical, and biological properties (Meng and Li, 2019). Under acidic conditions, this process induces fundamental changes in microbial community composition and enzyme activity critical for SOC stability (Rowley et al., 2020). In ecosystems characterized by N restriction, such as permafrost and peatland regions, N enrichment enhances N availability, accelerating the decomposition of labile organic C. This, in turn, results in decreased soil C availability (Craine et al., 2007; Janssens et al., 2010; Song et al., 2017). A previous study at our research site revealed a significant reduction in the soil labile C pool within the particulate organic C fraction with increasing N enrichment, signifying a decline in soil C availability (Chen et al., 2019). Our findings demonstrate that N enrichment significantly stimulates extracellular enzyme (EnC) activities and enhances microbial demand for C (Figure 2), aligning with prior research indicating that added N stimulates the activity of soil cellulose-degrading enzymes (e.g., cellobiosidase (CB) and β-glucosidase (BG)) (Carreiro et al., 2000; Saiya-Cork et al., 2002; Chen et al., 2017). This stimulation may be attributed to the increase in C-acquiring enzymes resulting from heightened microbial demand for C, especially in N-limited ecosystems (Keeler et al., 2009). Previous studies suggest that N enrichment could induce C limitation by reducing plant allocation to fine root production, leading to lower C input into the soil (Treseder, 2008). Thus, we propose that factors beyond the thermal environment, such as N enrichment, can modulate soil enzymes and alter substrate availability. Moreover, these processes can mediate the strength of the soil C-climate feedback. These results underscore the importance of considering soil C availability and enzymatic activity

responses, which collectively determine the response of the C balance to multiple environmental changes, for a more comprehensive understanding of C storage dynamics.'

L333-335: According to results of DRIFT and solid-state $^{13}$C NMR, warming did not have a significant effect on the soil organic matter composition, so what is meant by synchronous reduction here?
Simultaneous reduction refers to the size of the carbon pool.

L372: This study area is not permafrost soils, it is suggested to delete the sentence.
Done as suggested.

L737, L741, L754: What does "No significant differences were found" mean?
These sentences have been deleted.

L738: "Our"
Sorry for the mistake. It has been corrected.

L742: "The factors"
Sorry for the mistake. It has been corrected.

L757: There are three pictures, what do they represent? In addition, what do the different colors and pentagrams in the left figure represent?
The title of the Figure 1 has been rewritten to minimize incomprehension:
'**Figure 1**. Elevation map of the studied sites (a, the pentagram refers to the sampling point), photo of the alpine meadow (b) and the diagram of the warming treatment (open-top chamber) (c).'

L762: What do the abbreviations at the top right of the figure mean? Please add clarification. It is recommended that Figure (f) be formatted in the same way as the other figures.
Done as suggested. The abbreviations in the chart were included in the title page.

---

## Author Comment (AC3)

Dear Editor and Reviewer,

We extend our gratitude for your dedicated time invested in reviewing our manuscript and for sharing your invaluable insights. Your detailed feedback is deeply appreciated. Your constructive comments have significantly contributed to the enhancement of our work.

According to your comments, we have thoroughly revised the manuscript. We incorporated the necessary changes and refinements throughout the revised manuscript. To facilitate your review, we have provided detailed responses to each comment in blue color below.

Once again, we sincerely acknowledge the value of your time and your expert evaluation of our work. We hope that these revisions have fortified the manuscript's lucidity, precision, and overall quality.

Best Regards,

Sun Huimin on behalf of all co-authors.

**Response to RC#1**

This manuscript presents interesting evidence that long-term warming and N additions reduce soil carbon stocks, with significant interactions. However, the non-significant response of soil organic matter quality to the treatments was unexpected. The study has important implications for the fate of soil C in high-altitude areas where climate change is proceeding rapidly.

We greatly appreciate your comments and constructive suggestions regarding our study. Your comments and suggestions have been meticulously reviewed and integrated into the manuscript to improve its quality. Please refer to the responses for details.

The manuscript would benefit from inclusion of hypotheses regarding the changes in the SOC structure they expect to see in response to the warming and N addition treatments.

Thanks for this great suggestion. In the revision, we have added the hypotheses.

"We hypothesized that 9-years N enrichment and warming would affect SOC stock and the chemical structure of the SOC. N enrichment below a certain threshold may favor C sequestration in the alpine grassland ecosystem but warming may result in the C loss. And added N would stimulate hydrolytic enzyme activity while warming would repress enzyme activity. Finally, we hypothesized that variation in enzyme response to

N and temperature would emerge as an important explanation for variability in the effect of added N and warming on SOC stock."

The authors demonstrated significant interactions between the two global change treatments on many of their results, including aboveground biomass, soil pH, enzymes, C/N, SOC stocks. However, they have mostly ignored these interactive effects in reporting their results and interpreting them in the discussion section. Furthermore, they built the structural equation models without considering these interactions. Why? It seems like better insights into the multiple global change effects could be gained from establishing clear hypotheses (such as the a priori model) and then testing them with the SEM. The discussion and conclusion sections will need to be completely rewritten after considering the interacting global change effects more carefully.

This study focuses more on the changes brought about by nitrogen addition and warming on the carbon pool and its chemical molecular stability. The manuscript has been supplemented with a description of the results of the interactions after taking into account the comments of the two reviewers. And it is important given the focus of SEMs on explaining outcomes and not on identifying causative variables. The SEM model does not work well for the interactions, so to emphasize the importance of the enzyme activity we have only shown the results of nitrogen addition and warming. We hope to get your understanding.

Specific comments:

L 80-81, just give the background, remove the section that sounds like methods.

Thanks for your suggestion. The description about DRIFT has been moved to the METHOD section.

L 120-133, Site description section could be better organized – keep the plant community info together with the vegetation description. How large is the area of alpine grassland in Tibet? Give a comparison to other alpine grassland areas globally. How deep are the soils on average? Make sure to explain abbreviations like TP.

All done as suggested. The Site description section has been reorganized. Surface layer (0-10 cm) soils were collected in this study, see Experimental design section please. The description of the area of grassland ecosystem in the Tibetan Plateau has been added. The abbreviations have been explained well.

L 152, keep the verb tense consistent; past is preferred.

Done as suggested.

L 153, how was bulk density determined?

The METHOD has been added as follows. 'Bulk density samples were dried at 105 °C for 48 h and calculated by dividing the oven-dried soil mass by the steel cylinder volume (100 cm$^3$) because coarse fragments (stones or large roots) were not obtained in ring samples.'

L 193, please give a brief background in the intro on the connection between the hydrophobic/hydrophilic ration and soil aggregation stability, including how it relates to SOC stability.

Done as suggested. We added more explanations: 'The hydrophobic C/hydrophilic C (HB/HI) ratio, (C0–45 + C110–165)/(C45–110 + C165–210), was used to reflect the stability of soil aggregation (Spaccini et al. 2006, Wang et al. 2010). The higher values of HB/HI ratio indicated that SOC was more hydrophobic (Cao et al., 2016), which, in turn, implied that SOC was more stable (Spaccini et al., 2006, Wu et al., 2014).'

L 218, make sure to define all abbreviations on first use.

Done as suggested. Other parts of the text have also been scrutinized and corrected.

L 260, give year of citation

Done as suggested.

L 274 and L 325, you should not ignore the significant interaction between warming and N addition. L 282, the interaction that you observed between N and warming on AGB should also not be ignored.

Thanks for your suggestion. The content about AGB has been rewritten as follows:

'We noticed that warming significantly reduced AGB under N1 and N2 enrichments in this study (Figure 2).'

Besides, since warming and nitrogen addition have different effects on the organic carbon pool, the effect of nitrogen addition has been added to the discussion in a separate paragraph. See the following please.

'While N fertilization exerts both direct and indirect impacts on SOC, its influence on carbonates is direct, leading to continuous losses. This not only serves as a source of atmospheric CO2 (Kim et al., 2020; Raza et al., 2020; Zamanian et al., 2018) but also degrades soil structure and affects physical, chemical, and biological properties (Meng and Li, 2019). Under acidic conditions, this process induces fundamental changes in microbial community composition and enzyme activity critical for SOC stability (Rowley et al., 2020). In ecosystems characterized by N restriction, such as permafrost and peatland regions, N enrichment enhances N availability, accelerating the decomposition of labile organic C. This, in turn, results in decreased soil C availability

(Craine et al., 2007; Janssens et al., 2010; Song et al., 2017). A previous study at our research site revealed a significant reduction in the soil labile C pool within the particulate organic C fraction with increasing N enrichment, signifying a decline in soil C availability (Chen et al., 2019). Our findings demonstrate that N enrichment significantly stimulates extracellular enzyme (EnC) activities and enhances microbial demand for C (Figure 2), aligning with prior research indicating that added N stimulates the activity of soil cellulose-degrading enzymes (e.g., cellobiosidase (CB) and β-glucosidase (BG)) (Carreiro et al., 2000; Saiya-Cork et al., 2002; Chen et al., 2017). This stimulation may be attributed to the increase in C-acquiring enzymes resulting from heightened microbial demand for C, especially in N-limited ecosystems (Keeler et al., 2009). Previous studies suggest that N enrichment could induce C limitation by reducing plant allocation to fine root production, leading to lower C input into the soil (Treseder, 2008). Thus, we propose that factors beyond the thermal environment, such as N enrichment, can modulate soil enzymes and alter substrate availability. Moreover, these processes can mediate the strength of the soil C-climate feedback. These results underscore the importance of considering soil C availability and enzymatic activity responses, which collectively determine the response of the C balance to multiple environmental changes, for a more comprehensive understanding of C storage dynamics.'

L 355, try to write more succinctly – sentences like this contain multiple redundancies that obscure their meaning.

Done as suggested. The sentence has been rewritten in a simpler syntax to make it clearer: 'Microbial utilization of recalcitrant C pools could substantially accelerate overall soil C loss. This is because depolymerization of these recalcitrant macromolecules increases microbial accessibility to litter and SOC that was protected by recalcitrant C pools before (Schmidt et al. 2011, Lehmann and Kleber 2015, Paustian et al. 2016).'